# A pharmacist-led interprofessional medication adherence program improved adherence to oral anticancer therapies: The OpTAT randomized controlled trial

**Carole Bandiera**[1,2,3], **Evelina Cardoso**[4], **Isabella Locatelli**[3], **Khalil Zaman**[5], **Antonella Diciolla**[5], **Antonia Digklia**[5], **Athina Stravodimou**[5], **Valérie Cristina**[5], **Veronica Aedo-Lopez**[5], **Ana Dolcan**[5], **Apostolos Sarivalasis**[5], **Hasna Bouchaab**[5], **Jérôme Pasquier**[3], **Jennifer Dotta-Celio**[3], **Solange Peters**[5], **Dorothea Wagner**[5], **Chantal Csajka**[1,2,4], **Marie Paule Schneider**[1,2]*

1 School of Pharmaceutical Sciences, University of Geneva, University of Lausanne, Geneva, Switzerland,
2 Institute of Pharmaceutical Sciences of Western Switzerland, University of Geneva, Geneva, Switzerland,
3 Center for Primary Care and Public Health (Unisanté), University of Lausanne, Lausanne, Switzerland,
4 Center for Research and Innovation in Clinical Pharmaceutical Sciences, Lausanne University Hospital and University of Lausanne, Lausanne, Switzerland, 5 Department of Oncology, Lausanne University Hospital and University of Lausanne, Lausanne, Switzerland

* marie.schneider@unige.ch

**Data Availability Statement:** The data presented in this study are not publicly available due to ethical reasons, as imposed by the Ethics Committee

## Abstract

### Background

Oral anticancer therapies such as protein kinase inhibitors (PKIs) are increasingly prescribed in cancer care. We aimed to evaluate the impact of a pharmacist-led interprofessional medication adherence program (IMAP) on patient implementation (dosing history), persistence (time until premature cessation of the treatment) and adherence to 27 PKIs prescribed for various solid cancers, as well as the impact on patients' beliefs about medicines (BAM) and quality of life (QoL).

### Methods

Patients (n = 118) were randomized 1:1 into two arms. In the intervention arm, pharmacists supported patient adherence through monthly electronic and motivational feedback, including educational, behavioral and affective components, for 12 months. The control arm received standard care plus EM without intervention. All PKIs were delivered in electronic monitors (EMs). Medication implementation and adherence were compared between groups using generalized estimating equation models, in which relevant covariables were included; persistence was compared with Kaplan–Meier curves. Information on all treatment interruptions was compiled for the analysis. Questionnaires to evaluate BAM and QoL were completed among patients who refused and those who accepted to participate at inclusion, 6 and 12 months post-inclusion or at study exit.

"Commission cantonale d'éthique de la recherche sur l'être humain" (Vaud, Switzerland, +41213161836, scientifique.cer@vd.ch). Metadata and codebooks are available at the following address: https://doi.org/10.16909/DATASET/45. For any questions, please contact the documentation and data unit at the Center for Primary Care and Public Health Unisanté (Route de Berne 113, 1010 Lausanne, Switzerland, dfri. data@unisante.ch), through the data repository stated above.

**Funding:** The OpTAT study was funded by the Swiss Cancer Research Foundation, grant HSR-4077-11-2016 (MS). https://www.cancerresearch. ch/ The funders had no role in study design, data collection and analysis, decision to publish, or preparation of the manuscript.

**Competing interests:** The authors have declared that no competing interests exist.

**Abbreviations:** BAM, Beliefs about medicines; BMQ, Beliefs about medicines questionnaires; CI, Confidence interval; CML, Chronic myeloid leukemia; CONSORT, Consolidated standards of reporting trials; COVID, Coronavirus disease; CRF, Case report form; EM, Electronic monitor; EMERGE, ESPACOMP medication adherence reporting guidelines; EORTC-QLQ-C30, European organization for research and treatment of cancer quality of life questionnaire; GEE, Generalized estimating equation; GIST, Gastrointestinal stromal tumor; HCP, Health care providers; HER-2, Human Epidermal Receptor 2; IMAP, Interprofessional medication adherence program; IQR, Interquartile ranges; IV, Intravenous; LCD, Liquid crystal display; mTORi, mammalian target of rapamycin inhibitors; OAT, Oral anticancer therapy; OpTAT, Optimizing targeted anticancer therapies study; PKI, Protein kinase inhibitor; QoL, Quality of life; RS, Raw score; VEGF, Vascular Endothelial Growth Factor.

## Results

Day-by-day PKI implementation was consistently higher and statistically significant in the intervention arm (n = 58) than in the control arm (n = 60), with 98.1% and 95.0% (Δ3.1%, 95% confidence interval (CI) of the difference 2.5%; 3.7%) implementation at 6 months, respectively. The probabilities of persistence and adherence were not different between groups, and no difference was found between groups for BAM and QoL scores. No difference in BAM or QoL was found among patients who refused versus those who participated. The intervention benefited mostly men (at 6 months, Δ4.7%, 95% CI 3.4%; 6.0%), those younger than 60 years (Δ4.0%, 95% CI 3.1%; 4.9%), those who had initiated PKI more than 60 days ago before inclusion (Δ4.5%, 95% CI 3.6%; 5.4%), patients without metastasis (Δ4.5%, 95% CI 3.4%; 5.7%), those who were diagnosed with metastasis more than 2 years ago (Δ5.3%, 95% CI 4.3%; 6.4%) and those who had never used any adherence tool before inclusion (Δ3.8%, 95% CI 3.1%; 4.5%).

## Conclusions

The IMAP, led by pharmacists in the context of an interprofessional collaborative practice, supported adherence, specifically implementation, to PKIs among patients with solid cancers. To manage adverse drug events, PKI transient interruptions are often mandated as part of a strategy for treatment and adherence optimization according to guidelines. Implementation of longer-term medication adherence interventions in the daily clinic may contribute to the improvement of progression-free survival.

## Trial registration

ClinicalTrials.gov NCT04484064.

## I. Introduction

### 1. Background

A total of 19.3 million people were diagnosed with cancer worldwide in 2020, and 10 million patients died from this disease [1]. Despite an increasing incidence of cancer cases, mortality is decreasing in developed countries, thanks to early diagnosis and cancer treatments, including oral anticancer therapies (OATs) [2] and immunotherapy. OATs include a large variety of therapeutic agents, such as cytotoxic and immunomodulatory drugs, hormonal antagonists, and targeted agents that complement or substitute intravenous (IV) chemotherapy. Targeted agents include protein kinase inhibitors (PKIs), which are inhibitors of mutated or overexpressed protein kinases. These kinases modulate oncogenic signaling, leading to uncontrolled cancer cell growth and invasion [3]. In 2001, the PKI imatinib was launched in the pharmaceutical market and revolutionized cancer care by showing significant clinical benefit and prolonged survival among patients with chronic myeloid leukemia (CML) [4] and gastrointestinal stromal tumor (GIST). Since then, an increasing number of OATs have been marketed for a wide variety of cancers, and as many as 11 compounds were approved by the United States Food and Drug Administration (FDA) in 2020 [5]. Beyond the established evidence of improved outcomes, patients often prefer oral administration route, such as OAT compared to IV chemotherapy, because it is more convenient and less

invasive, and no visit to a hospital center is required [6]. Although OATs increase patient autonomy, patient responsibility in self-managing the treatment at home is extensive, and the oncology health care team cannot directly supervise medication adherence and acute adverse events management.

Medication adherence is defined by the extent to which a patient takes the treatment as prescribed, ideally following a shared decision-making process with the health care team. Medication adherence is characterized by three interrelated and quantifiable phases: treatment initiation (i.e., first dose taken), treatment implementation (i.e., the extent to which the patient's dosing history corresponds to the prescription according to the correct dosing regimen, the correct timing and other specific requirements) and discontinuation (i.e., the patient stops taking the treatment prematurely) [7]. Persistence in treatment is the time between the first and the last dose taken [7]. Achieving optimal adherence is paramount to reach targeted clinical outcomes [8], as nonadherence to OATs increases drug resistance, leads to treatment failure [9, 10] and decreases survival [11–13]. According to the literature, it is estimated that one-quarter to one-third of patients do not implement or persist with OATs [14, 15] due to a large variety of clinical, personal and contextual determinants (i.e., adverse events, toxicity, regimen complexity, low perceived need for OATs at a distance from diagnosis, forgetfulness, and high out-of-pocket costs in some countries) [16–18]. To address these determinants, interventions to improve adherence to OATs have been increasingly reported in the last decade, but their evaluation and evidence on the improvement of adherence and clinical outcomes remain limited. However, multifactorial interventions tailored to patients' needs (i.e., education, counseling, behavioral interventions), such as pharmacist-led programs, have shown promising results [19–22]. While many studies have monitored adherence to endocrine therapies among patients with breast cancer or adherence to PKIs in CML, little is known about adherence to other PKIs prescribed for solid tumors. There is a particular gap in the literature on adherence to palliative treatment lines in solid cancers. The Interprofessional Medication Adherence Program (IMAP), implemented for almost 30 years at the community pharmacy of the Center of Primary Care and Public Health *Unisanté* (Lausanne, Switzerland) [23, 24], supports adherence and retention in care for patients with long-term conditions [25, 26]. In a prior study, even if medication implementation was high and stable among 43 patients included in the IMAP treated with endocrine therapies and cytotoxic agents, 15% had discontinued their medication at 12 months [22]. In this context, the randomized and controlled Optimizing Targeted Anticancer Therapies (OpTAT) study was implemented at the community pharmacy of *Unisanté* and at Lausanne University Hospital, located in the same hospital complex.

## 2. Objectives

The first objective of the OpTAT study was to evaluate the longitudinal impact of a pharmacist-led IMAP (= intervention group) on implementation, persistence and adherence to PKIs over 12 months compared to the standard of care (= control group). The second objective was to evaluate the impact of patients' clinical and demographic covariables on the implementation of PKIs in both groups. The third objective was to describe patient implementation longitudinally by considering prescribed PKI transient interruptions compared to the usual recommended regimen by pharmaceutical industries. Fourth, we aimed to evaluate the impact of the IMAP on patients' beliefs about medicines (BAM) and their quality of life (QoL) at study inclusion and at 6 and 12 months and to compare BAM and QoL between patients who accepted and those who refused to participate.

## 3. Outcomes

We considered a daily medication intake outcome, defined as a longitudinal binary variable (correct intake = 1; incorrect intake = 0) measured for each patient on each day of the monitoring period. The medication intake was considered correct (= 1) on a given day $t$ when the patient took at least all the prescribed drug doses for each electronic monitor (EM) that day and incorrect (= 0) otherwise. Empirical implementation was defined on each day $t$ by the proportion of patients with a correct medication intake (proportion of outcomes = 1) among patients still participating in the study that day [22].

OAT persistence was a second continuous outcome, defined as the individual time between study inclusion and treatment discontinuation.

Empirical adherence was defined on each day $t$ by the proportion of patients with correct medication intake among all patients initially included in this study, including patients who discontinued medication before time $t$.

## 4. Hypothesis

We hypothesized that PKI implementation and adherence among patients included in the intervention group (i.e., benefiting from the IMAP) would be improved and would stay stable over time compared to control patients, and that patients in the intervention group would persist longer on PKIs. We further hypothesized that improvements in adherence would also result in a better QoL of patients in the intervention group, and that the intervention would influence patients' beliefs and reduce patients' concerns about taking PKIs. We hypothesized that PKI implementation would be affected by demographic (e.g., age, gender) and clinical determinants (e.g., time since PKI initiation, presence of distant metastases, time since diagnosis of distant metastases) and previous use of adherence tools. We hypothesized that considering PKIs alternate regimens (i.e., transient interruptions of PKI) in the implementation calculation, rather than the usual regimens recommended by the pharmaceutical industry, would allow us to highlight the risk of underestimating the implementation outcome, when researchers do not consider the alternative regimens that often occur in routine care.

## II. Methods

### 1. Ethical considerations and guidelines

The OpTAT study was approved by the local ethics committee "*Commission cantonale d'éthique de la recherche sur l'être humain*" (Vaud, Switzerland, ID 65/15) in 2015. All included patients signed an informed written consent form to participate. This study was conducted in accordance with the Declaration of Helsinki. Both the ESPACOMP Medication Adherence Reporting Guidelines (EMERGE) [27] and Consolidated Standards of Reporting Trials (CONSORT) guidelines [28] were used to report findings.

### 2. The OpTAT medication adherence study

**Design of the OpTAT study.** The OpTAT protocol has been published elsewhere [29]. Briefly, the OpTAT study is monocentric, open, and composed of two parts: i) a randomized controlled medication adherence study and ii) a combined PKI pharmacokinetic and pharmacodynamic analysis based on the collection of patient blood samples. While partial results were reported in a subgroup of patients included in OpTAT and treated with palbociclib—a PKI prescribed for metastatic breast cancer [30]—, this paper presents the results of the medication adherence part for all patients included in OpTAT.

Patients were recruited at the Department of Oncology of Lausanne University Hospital. The first patient was included on July 24th, 2015, and data were collected until the last visit of the last included patient on May 3rd, 2022. The sample size calculation (n = 120 patients, 60 in each group) is presented elsewhere [29]. Eligible patients were adults treated with at least one oral PKI for solid cancers. Patients were excluded if they did not self-manage their treatment (i.e., benefited from home care services or caregivers or were under tutelage) or if they were diagnosed with major cognitive impairments. Patients' reasons for refusal to participate in this study were collected in case report forms (CRFs).

Each PKI was delivered in an EM (Medication Event Monitoring System, MEMS and MEMS AS, AARDEX Group, Sion, Switzerland) for all included patients. The EM is an interactive digital technology composed of a pill bottle and a cap, in which a chip records the time and date of each EM opening and is considered a proxy for drug intake. A liquid crystal display (LCD) screen on top of the EM cap informed the patient about the number of daily EM openings (i.e., from 3:00 am to 2:59 am the next day). Baseline medication adherence was monitored by EM for at least 21 days, after which patients were randomized 1:1 in two parallel arms (intervention or control) (see S1 Appendix, adapted from Bandiera et al. [29]). To equitably distribute cancer types and PKI experience between groups, randomization was stratified per cancer type and time between PKI initiation and study inclusion (i.e., more or less than 30 days). The randomization sheet was created by Excel (Microsoft, version 2016) based on variable size block randomization and listed 1 (intervention group) and 0 (control group). The sheet was provided by an independent researcher from the Unisanté Research Support Unit [29].

**Intervention group: Use of the EM as part of the IMAP.** The IMAP consisted of a monthly 15-minute face-to-face patient-pharmacist interview conducted in an interview room; medication adherence feedback was provided to the patient using the spirit and techniques of motivational interviewing, guided by the theoretical model of Fisher et al. ("information-motivation-behavioral skills") [31].

Medication adherence feedback was provided based on sharing the EM chronology plot with the patient (i.e., a graph representing the day-by-day occurrence and timing of each EM opening since the last interview). Importantly, before showing the chronology plot as feedback to the patient, the pharmacist asked patients to report i) the estimated perceived number of missed doses since the last interview; ii) the nonmonitored periods during which the EM was not used (e.g., during hospitalizations); iii) and the use of pocket doses (i.e., when the patient opened the EM in advance to take the dose(s) the day after during which the EM was not opened) or curiosity checks (i.e., EM openings without drug intake). In addition, pharmacy technicians performed a pill count to calculate an aggregated percentage of medication intake since the last refill. The pill count allowed pharmacists to reconcile EM and pill count data to strengthen the methodology, especially in cases of multiple tablets per intake.

In a nonjudgmental manner, the pharmacist investigated the patient's needs in terms of information on medication and medication behavior and the patient's motivation and readiness to take the treatment and explored the patient's daily medication and adverse event management. If medication adherence was suboptimal, the pharmacist explored the patient's willingness and capacity to change his or her behavior. BAM and QoL were also explored and addressed whenever needed.

After the intervention, the pharmacist sent a structured adherence report summarizing the content of the intervention to the health care team (e.g., oncologist, nurses) as a guidance and reinforcement tool for asynchronous interprofessional collaborations. Notably, during the coronavirus disease 2019 (COVID-19) pandemic lockdown, interviews were conducted by phone, and pill boxes were sent by mail so that patients could refill their EM at home [32].

**Control group: Use of the EM in addition to the standard of care.** Patients included in the control group used the EM but did not benefit from the IMAP. EM adherence data were blinded to the patient, the pharmacy, and the clinical and research teams. The patient came to the pharmacy after each oncology consultation to fill the EM with the prescribed PKI. At each pharmacy visit, the pharmacist asked predefined questions to the patient at the pharmacy counter to report any deviation from the expected EM use (e.g., pocket doses, nonmonitored periods), which were reported in a CRF. The pharmacy technicians counted the number of pills returned to the pharmacy without calculating the aggregated percentage of medication intake.

**Questionnaires.** In both groups, patients were asked to complete a set of questionnaires validated in French with good psychometric properties at inclusion and 6 and 12 months after inclusion: i) the validated French version of the Beliefs about Medicines Questionnaire (BMQ) [33] developed by Horne et al. [34], which evaluates perceived necessity, beliefs, concerns and prejudices about the treatment, and ii) the validated French version of the European Organization for Research and Treatment of Cancer Quality of Life Questionnaire (EORTC-QLQ-C30, version 3.0) [35], which evaluates QoL through patient functioning, symptoms and global health status. Patients who refused to participate in this study were also invited to complete both questionnaires anonymously.

*i) BMQ.* The BMQ is composed of 18 questions rated on a 5-point Likert scale from 1 "Strongly agree" to 5 "Strongly disagree". The questionnaire evaluated four scales of BAM: specific beliefs, including the perceived necessity for the treatment (5 questions) and concerns about the treatment (5 questions), and general beliefs, including the perception of overprescribing (4 questions) and perceived prejudices of the treatment (4 questions). For each scale, a total score was calculated by adding the reverse scores of the questions, which ranged from 5 to 25 for perceived necessity and concerns and from 4 to 20 for perceived overprescribing and prejudices [33]. Higher scores indicated stronger beliefs [33].

*ii) EORTC-QLQ-C30.* The EORTC-QLQ-C30 is composed of 30 questions and explores patients' functional status (15 questions) and patients' symptoms (13 questions), rated on a 4-point Likert scale from 1 "Not at all" to 4 "Very much" and perceived global health status (2 questions) rated on a 7-point Likert scale from 1 "Very poor" to 7 "Excellent". Each scale was scored as follows: the mean of the score per question (raw score, RS) was linearly transformed to a 0–100 score (S) as defined by the guidelines (i.e., for the functional scale, $S = (1-(RS-1)/3) \times 100$; for the symptom scale, $S = ((RS-1)/3) \times 100$ and for the global health status, $S = ((RS-1)/6) \times 100$ [35, 36]. A high score for the functional scale indicated a healthy level of functioning, a high score for the global health status indicated high QoL, and a high score on the symptom scale indicated a high level of symptomatology [36].

## 3. Database construction

**Collection of patients' clinical and sociodemographic data and questionnaires.** At study inclusion, patients' sociodemographic (i.e., age, gender, civil status, ethnicity) and clinical data (i.e., cancer type, time since primary and metastatic cancer diagnoses, presence of distant metastases, PKI treatment objective (if palliative, number of palliative lines), time since PKI initiation, actual oncology treatment (name of actual PKI, number of PKIs monitored and combined anticancer treatments), previous oncology treatments, number of oral prescribed chronic nononcologic treatments, and use of adherence tools (if yes, adherence tools used) were retrieved from the electronic medical and administrative records (Soarian, Oracle Cerner, USA) of Lausanne University Hospital.

Patient-reported data regarding the study follow-up were collected in CRFs (i.e., reasons for drop-out, attendance to the IMAP after the end of the study). Patients completed the BMQ and EORTC-QLQ-C30 on paper. All data were gathered in the secure web platform REDCap™ version 6.13.3 (Vanderbilt University) [37].

**EM adherence database.** The raw EM database was cleaned and enriched using a semiautomated procedure developed in the statistical software R by our research team (i.e., CleanADHdata.R script available on https://github.com/jpasquier/CleanADHdata). For each patient, we truncated the EM adherence database by the actual start and end dates of EM use, and we adapted the number of expected EM openings based on the prescription sheets; we considered all adaptations over time (i.e., dosage and regimen changes and transient interruptions) by exploring and reconciling all available sources: patient medical records, pharmacy records and patient reports. In the case of EM nonopenings, if a patient reported pocket doses, confirmed by pill count, we corrected the number of EM nonopenings by inserting the number of pockets doses reported by the patient. We introduced the nonmonitored periods as collected in the CRFs, for which the implementation outcome was missing. We enriched the EM adherence database with patient covariables (e.g., age, group of randomization, presence of treatment discontinuation), EM covariables (e.g., medication international nonproprietary name, dosing strengths) and adverse events reported by patients at each medical visit.

To answer our first and second objectives, in the case of transient interruptions in PKIs prescribed for (a) clinical reasons for treatment optimization according to guidelines (e.g., toxicity, concomitant treatments, infections [30]) by the oncologist (i.e., considered alternate regimens), (b) administrative reasons (e.g., medical appointment postponed, PET scan results pending [30]) or (c) patient requests validated in advance with the prescriber (e.g., holidays [30]), the number of expected EM openings was 0, categorized as optimal implementation according to our definition. In contrast, to answer our third objective, all EM expected openings during the transient interruptions in PKIs prescribed by oncologists were compared to the usual regimen recommendation provided by pharmaceutical industries.

## 4. Statistical analysis

**a. Descriptive analysis.** All analyses were performed by original assigned groups (i.e., intention-to-treat principle). In both groups, sociodemographic and clinical variables are presented as proportions for categorical variables and as medians and interquartile ranges (IQRs) for continuous variables.

**b. Implementation.** For each EM assigned to each patient, the medication intake was considered correct (= 1) if the number of observed EM openings was at least equal to the number of expected EM openings based on the prescription of the oncologist; medication intake was considered incorrect otherwise (= 0). On each day of the monitoring period, empirical implementation was defined by the proportion of patients with correct medication intake (proportion of outcomes = 1) among patients still participating in this study that day [22]. For example, implementation on day d is x/y = z%, since x out of y patients still under observation on day d were taking their medication according to their prescription.

From study inclusion to 12 months after inclusion, longitudinal implementation was then described by a generalized estimating equation (GEE) model on daily medication intake 0/1, with an autoregressive correlation structure and a polynomial time effect.

Since patients experienced a baseline period before being randomized either to stay in the control group or to be included in the intervention group, the group variable was introduced in the GEE model with two time-dependent variables representing on each day $t$ the time spent in the intervention and the time spent in the control group until $t$. The implementation

model estimate was presented at 6 months after inclusion (i.e., when the impact of the intervention could be best evaluated, e.g., with most patients still participating in this study) for a representative patient who remained in the control group after randomization and for a representative patient who switched to the intervention group after the baseline period of at least 21 days. The difference (Δ) in treatment implementation and the 95% confidence intervals (95% CIs) of the difference between the two representative patients at 6 months are also presented.

Patient age, gender, use of adherence tools, time between PKI initiation and study inclusion, time between metastatic diagnosis and study inclusion, and the presence of distant metastases were included as covariables one at a time in the GEE model as an exploratory analysis. Continuous variables were dichotomized at their median value.

**c. Persistence and adherence.**   We defined a PKI discontinuation as a premature PKI cessation for adverse events (i.e., symptoms experienced by the patient), based (or not) on shared decision-making between the patient and the oncologist, or for any other patient personal or unilateral reasons. Other reasons for premature PKI cessation (i.e., clinical reasons other than adverse events, cancer progression or toxicity for which no symptoms were experienced by the patient) or study interruptions without discontinuing PKIs were considered censoring times. Treatment persistence was characterized by the distribution of the continuous outcome defined as the individual times between study inclusion and treatment discontinuation, and it was estimated by Kaplan–Meier survival curves, which allows to account for censored durations [22].

Empirical adherence was defined on each day by the proportion of patients with a correct medication intake (outcome = 1) among all patients initially included in this study [22], corresponding to the product between the probabilities of PKI implementation and persistence on each day of the monitoring period [38]. Adherence was then compared between study groups using the GEE model, using a predefined methodology [38]. The baseline period was excluded from this analysis (i.e., data were included from the randomization date), and the monitoring period was considered until 340 days (i.e., approximately 12 months minus the 21 days of the baseline period). The adherence model estimate was presented at 6 months postrandomization in both groups along with the difference between groups with 95% CIs of the difference.

**d. Questionnaires.**   For each scale of both the BMQ and the EORTC-QLQ-C30, the mean scores and their standard deviations were reported for patients who refused to participate but completed the questionnaire at enrollment and for patients who agreed to participate and completed the questionnaire at study inclusion. The mean scores and their standard deviations were also reported for patients included in the intervention and control groups at 6 months and 12 months post-inclusion. Welch's t tests were performed to compare scores between groups; a statistically significant difference was considered if p<0.05. Missing values were clearly depicted.

## III. Results

### 1. Included patients

A total of 241 eligible patients were identified, of whom 111 refused to participate. The main reasons for nonparticipation were collected for 103 patients and are presented in S2 Appendix. In total, 130 patients were included, of whom 12 left the study during the baseline period. Among them, 3 patients did not use the EM. The median time spent in the adherence study after inclusion was 184 (IQR 108; 359) days among patients included in the intervention group versus 356 days (IQR 176; 388) among patients in the control group (excluding nonrandomized patients). Notably, after study completion, 5 patients in the intervention group decided

to continue attending the IMAP, and 4 patients in the control group decided to start attending the IMAP.

EM data from 127/130 (97.7%) patients were analyzed. The sociodemographic and clinical variables of these 127 patients (intervention group n = 58, control group n = 60; not randomized n = 9) are presented in Table 1. In total, 27 PKIs or associations of PKIs were monitored, of which 4 PKIs were prescribed as a cyclic regimen. Most patients were diagnosed with gastrointestinal cancer or breast cancer. The majority of patients were diagnosed with metastasis, and PKIs were mainly prescribed with a palliative objective. Even if patients were not polymedicated (i.e., less than 5 chronic daily treatments prescribed [39]), approximately a quarter of the patients in both groups had used adherence tools in their therapeutic itinerary, which was in most cases a weekly pill box.

Face-to-face interviews with patients in the intervention group lasted a median of 18 minutes (IQR 11; 25), and pharmacists wrote the adherence report in 20 minutes (IQR 15; 25). The pharmacist met control patients during a median time of 5 minutes (IQR 5; 10), and the pharmacists took another 5 minutes (IQR 2; 5) to complete the CRF.

Fig 1 describes patient enrollment from inclusion to data analysis. In the intervention group, 15/58 (26%) patients completed the 12-month study. Among patients who discontinued the study, 35/43 (81%) patients left the study due to clinical reasons, and 8/43 (19%) patients left the study for personal reasons other than clinical. In the control group, 30/60 (50%) patients completed the 12-month study, and among patients who discontinued the study, 25/30 (83%) left the study due to clinical reasons, and 5/30 (17%) patients left the study for personal reasons other than clinical. In both groups, the main clinical reasons for premature PKI cessation were cancer progression, adverse events or toxicity. Two patients in the control group died during this study.

## 2. PKI implementation

PKI implementation is presented in Fig 2 for a representative patient randomized in the intervention group on day 21 (red line) versus a representative patient who stayed in the control group after randomization (blue line). Implementation of PKI improved among patients included in the intervention group since the first interview at 21 days and was constantly higher (i.e., at each single day) and more stable during the 12-month monitoring period compared to control patients. At 6 months, the estimation of the probability of PKI implementation in the intervention versus the control group was 98.1% and 95.0%, respectively (Δ3.1%, 95% CI of the difference 2.5%; 3.7%). Fig 3 shows the impact of covariables on PKI implementation in both groups. When comparing implementation in the intervention and the control groups, implementation was lower in men (at 6 months, Δ4.7%, 95% CI 3.4%; 6.0%), those aged less than 60 years old at study inclusion (Δ4.0%, 95% CI 3.1%; 4.9%), those who had initiated PKI more than 60 days ago at study inclusion (Δ4.5%, 95% CI 3.6%; 5.4%), patients who did not have distant metastasis at the time of study inclusion (Δ4.5%, 95% CI 3.4%; 5. %7), those who were diagnosed with distant metastasis more than 2 years before study inclusion (Δ5.3%, 95% CI 4.3%; 6.4%) and those who had never used any adherence tool before inclusion (Δ3.8%, 95% CI 3.1%; 4.5%). We conducted a sensitivity analysis exploring the impact of gender on PKI implementation by excluding patients with female breast cancers and gynecological cancers (n = 28 in the control and nonrandomized groups, n = 24 in the intervention group). Similar results were found compared to the analysis on the whole sample, with men implementing PKI less than women (Δ4.15, 95% CI 2.8%; 5.5%).

Table 2 shows the estimations of PKI implementation in both groups by the GEE models and the difference in implementation between groups and their 95% CI.

**Table 1. Patients' demographic and clinical data at study inclusion.**

| | Intervention (n = 58) | Control (n = 60) + not randomized (n = 9)[d] |
|---|---|---|
| **Demographic data at inclusion** | | |
| **Age (years), median (IQR)** | 61 (53.0; 70.7) | 61 (52.2; 68.4) |
| **Female gender, n patients (%)** | 37 (63.8) | 38 (55.1) |
| **Married patients[a], n patients (%)** | 30 (51.7) | 40 (58.0) |
| **Caucasian, n patients (%)** | 55 (94.8) | 61 (88.4) |
| **Clinical and pharmaceutical data at inclusion** | | |
| **Cancer type, n patients (%)** | Breast cancers 21 (36.2) Gastrointestinal cancers 17 (29.3) Melanoma 5 (8.6) Sarcoma 4 (6.9) Urologic cancers 4 (6.9) Oral cancers 3 (5.2) Gynaecologic cancers 3 (5.2) Lung cancers 1 (1.7) | Breast cancers 21 (30.4) Gastrointestinal cancers 22 (31.9) Melanoma 6 (8.7) Sarcoma 5 (7.3) Urologic cancers 4 (5.8) Oral cancers 1 (1.4) Gynaecologic cancers 6 (8.7) Lung cancers 4 (5.8) |
| **Time since primary cancer diagnosis (years), median (IQR)** | 4.2 (1.7; 8.0) Missing data n = 2 | 2.1 (0.9; 5.8) |
| **Presence of distant metastases (stage 4), n patients (%)** | 50 (86.2) | 53 (76.8) |
| **Time since metastatic diagnosis (years) among patients with metastasis, median (IQR)** | 1.9 (0.8; 2.9) Patients without metastases n = 8 | 1.6 (0.6; 2.4) Patients without metastases = 16 |
| **Objective of the PKI, n patients (%)** | Palliative 46 (79.3) Adjuvant 8 (13.8) Neo-adjuvant 4 (6.0) | Palliative 58 (84.1) Adjuvant 2 (2.9) Neo-adjuvant 9 (13.0) |
| **If palliative objective, palliative line, n patients (%)** | 1st line 19 (41.3) ≥3rd line 17 (37.0) 2nd line 10 (21.7) | 1st line 24 (41.4) ≥ 3rd line 22 (37.9) 2nd line 12 (20.7) |
| **New users[b], n patients (%)** | 5 (8.6) | 6 (8.7) |
| **Time (days) between initiation of monitored PKI and study enrolment, median (IQR)** | 57 (18; 207) | 63 (20; 177) |
| **Monitored anticancer molecule, n patients (%)[c]** | Palbociclib 19 (32.8) Regorafenib 7 (12.1) Pazopanib 6 (10.3) Imatinib 5 (8.6) Sorafenib 4 (6.9) Everolimus 3 (5.2) Lenvatinib 3 (5.2) Trametinib/Dabrafenib 3 (5.2) Axitinib 2 (3.4) Olaparib 2 (3.4) Cobimetinib 1 (1.7) Erdafitinib 1 (1.7) Erlotinib 1 (1.7) Niraparib 1 (1.7) Trametinib 1 (1.7) Trametinib/olaparib 1 (1.7) Vemurafenib 1 (1.7) Cabozantinib 1 (1.7) Binimetinib 0 (0) Alectinib 0 (0) Cobimetinib/Vemurafenib 0 (0) Encorafenib 0 (0) Lapatinib 0 (0) Osimertinib 0 (0) Ribociclib 0 (0) Sunitinib 0 (0) Neratinib 0 (0) | Palbociclib 18 (26.1) Regorafenib 4 (5.8) Pazopanib 4 (5.8) Imatinib 14 (20.3) Sorafenib 3 (4.3) Everolimus 2 (2.9) Lenvatinib 1 (1.4) Trametinib/Dabrafenib 5 (7.2) Axitinib 3 (4.3) Olaparib 2 (2.9) Cobimetinib 0 (0) Erdafitinib 0 (0) Erlotinib 0 (0) Niraparib 2 (2.9) Trametinib 1 (1.4) Trametinib/olaparib 0 (0) Vemurafenib 0 (0) Cabozantinib 0 (0) Binimetinib 1 (1.4) Alectinib 2 (2.9) Cobimetinib/Vemurafenib 1 (1.4) Encorafenib 1 (1.4) Lapatinib 1 (1.4) Osimertinib 1 (1.4) Ribociclib 1 (1.4) Sunitinib 3 (4.3) Neratinib 1 (1.4) |
| **Number of monitored PKI per patient, median (IQR)** | 1 (1; 1) | 1 (1; 1) |

(*Continued*)

**Table 1.** (Continued)

|  | Intervention (n = 58) | Control (n = 60) + not randomized (n = 9)[d] |
|---|---|---|
| **Previous oncologic treatments since cancer diagnosis, n patients (%)** | Tumor surgery 44 (75.9) <br> IV Chemotherapy 30 (51.7) <br> Radiotherapy or radiofrequency 29 (50.0) <br> Endocrine therapy oral 19 (32.8) <br> Immunotherapy 10 (17.2) <br> Anti-VEGF (Bevacizumab) 8 (13.8) <br> Oral chemotherapy (capecitabin) 6 (10.3) <br> Oral PKI other than mTORi 5 (8.6) <br> Endocrine IM (Fulvestrant) 5 (8.6) <br> Goserelin or leuprorelin 4 (6.9) <br> mTORi (everolimus) 3 (5.2) <br> Radioembolisation 3 (5.2) <br> Chemoembolisation 3 (5.2) <br> Cryoablation 2 (3.4) <br> Thermal ablation 1 (1.7) <br> Radiosurgery 0 (0) <br> Anti-HER-2 (trastuzumab) 0 (0) <br> No previous anticancer treatments n = 4 (6.9) | Tumor surgery 43 (62.3) <br> IV Chemotherapy 29 (42.0) <br> Radiotherapy or radiofrequency 28 (40.6) <br> Endocrine therapy oral 19 (27.5) <br> Immunotherapy 10 (14.5) <br> Anti-VEGF (Bevacizumab) 13 (18.8) <br> Oral chemotherapy (capecitabin) 6 (8.7) <br> Oral PKI other than mTORi 8 (11.6) <br> Endocrine IM (Fulvestrant) 3 (4.3) <br> Goserelin or leuprorelin 1 (1.4) <br> mTORi(everolimus) 2 (2.9) <br> Radioembolisation 1 (1.4) <br> Chemioembolisation 2 (2.9) <br> Cryoablation 2 (2.9) <br> Thermal ablation 1 (1.4) <br> Radiosurgery 1 (1.4) <br> Anti-HER-2 (trastuzumab) 3 (4.3) <br> No previous anticancer treatments n = 13 (18.8) |
| **Combined anticancer treatment in addition to PKI, n patients (%)** | Endocrine IM (Fulvestrant) 11 (19.0) <br> Endocrine therapy oral 9 (15.5) <br> IV Chemotherapy 1 (1.7) <br> Radiotherapy 2 (3.4) <br> Goserelin or leuprorelin 2 (3.4) <br> Anti-VEGF (Bevacizumab) 1 (1.7) <br> Immunotherapy 0 (0) <br> No concomitant anticancer treatment n = 45 (77.6) | Endocrine IM (Fulvestrant) 12 (17.4) <br> Endocrine therapy oral 8 (11.6) <br> IV Chemotherapy 0 (0) <br> Radiotherapy 2 (2.9) <br> Goserelin or leuprorelin 2 (2.9) <br> Anti-VEGF (Bevacizumab) 0 (0) <br> Immunotherapy 2 (2.9) <br> No concomitant anticancer treatment n = 35 (50.7) |
| **Number of oral prescribed chronic non-oncologic treatments, median (IQR)** | 3 (1; 5) | 3 (1; 5) |
| **Patients having used previous adherence supporting tools, n patients (%)** | 14 (25.0) <br> Missing data n = 2 | 17 (25.0) <br> Missing data n = 1 |
| **Adherence personal tools used, n patients (%)** | Weekly pill-box 13 (22.4) <br> Electronic monitor 0 (0) <br> Other personal tools 1 (1.7) | Weekly pill-box 16 (23.2) <br> Electronic monitor 1 (1.4) <br> Other personal tools 0 (0) |

NB: PKI = protein kinase inhibitors, IQR = interquartile ranges, IM = intramuscular, IV = intravenous, VEGF = Vascular Endothelial Growth Factor,

mTORi = mammalian target of rapamycin inhibitors, HER-2 = Human Epidermal Receptor 2.

[a] The other patients are divorced, single, widowed, separated or other;

[b] Patients considered as new users initiated their PKI ≤ 14 days before study inclusion [40];

[c] Some patients were monitored with more than one PKI;

[d] The nine patients, who were not randomized, did not benefit from the intervention, as they left the study during the baseline period. Their implementation outcomes contributed to implementation estimate during the baseline period. They are treated in the model as patients staying in the control group before being lost to follow-up.

## 3. Persistence and adherence to PKI

In total, 7 patients in the intervention group and 6 in the control group discontinued the treatment due to adverse events. S3 Appendix shows the Kaplan–Meier survival curves for treatment persistence during the 12-month monitoring period, as well as the estimation of the probability of PKI adherence by the GEE model. At 6 months, the probabilities for treatment persistence and adherence were comparable between groups: in the intervention and control groups, persistence was 91.5% and 89.3% (Δ2.3%, 95% CI -9.98; 14.50%), and adherence was 88.9% and 86.0% (Δ 2.8, 95% CI -10.06; 15.02%), respectively.

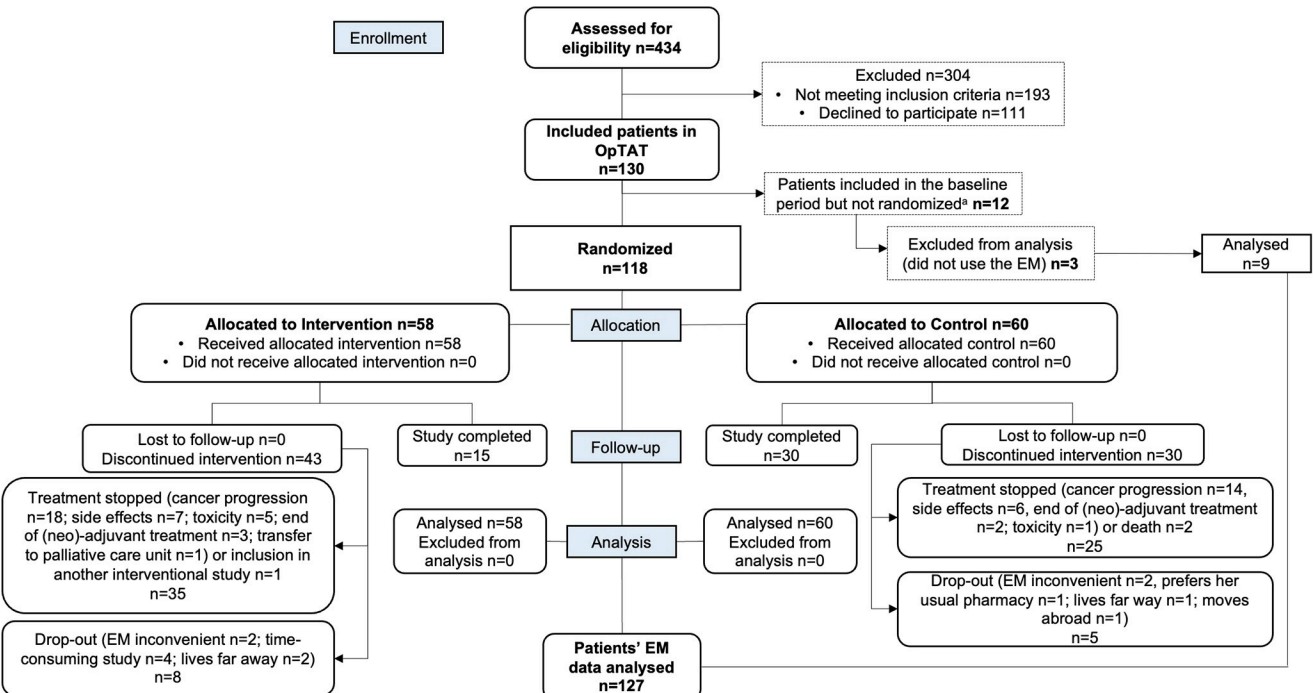

**Fig 1. Flow of patient enrollment from inclusion to data analysis.** NB: [a]patients left the study during the baseline period before randomization.

## 4. Comparative analysis of PKI implementation between the prescription of the alternate PKI regimens by oncologists and the usual recommended regimen by pharmaceutical industries

In total, 34/58 (58.6%) patients in the intervention group, 32/60 (53.3%) patients in the control group and 0/9 (0%) nonrandomized patients experienced at least one PKI transient interruption for more than 2 days in a row with at least one of their EMs. In the intervention and control groups, the median number of transient PKI interruptions of more than 2 days in a row per EM was 1 (IQR 0; 1), and the median number of days of each PKI transient interruption was 7 days (IQR 4; 12) and 7 days (IQR 5; 13), respectively.

Fig 4 shows the implementation of PKI considering i) the PKI regimen as prescribed, including alternate regimens for treatment optimization according to guidelines or for administrative reasons or on patient request, and ii) the usual regimen recommendation provided by pharmaceutical industries. Patient implementation of PKI at 6 months would have been lower (-7.8%) if the transient interruptions prescribed by the oncologists during the study were not considered in the analysis: 88.5% (95% CI 84.2%; 91.8%) vs. 96.3% (95% CI 94.5%; 97.5%).

## 5. BMQ and EORTC-QLQ-C30

The scores for BAM (i.e., perceived treatment necessity, concerns, prejudices and overprescribing) and for QoL (i.e., in terms of functioning, symptoms and global health status) were comparable in every dimension of the questionnaires between groups at 6 and 12 months post-inclusion (S4 Appendix). Moreover, no difference was found in both questionnaires between patients who refused and those who agreed to participate. Notably, scores for perceived concerns, prejudices and overprescribing were in the upper half of the scale for patients

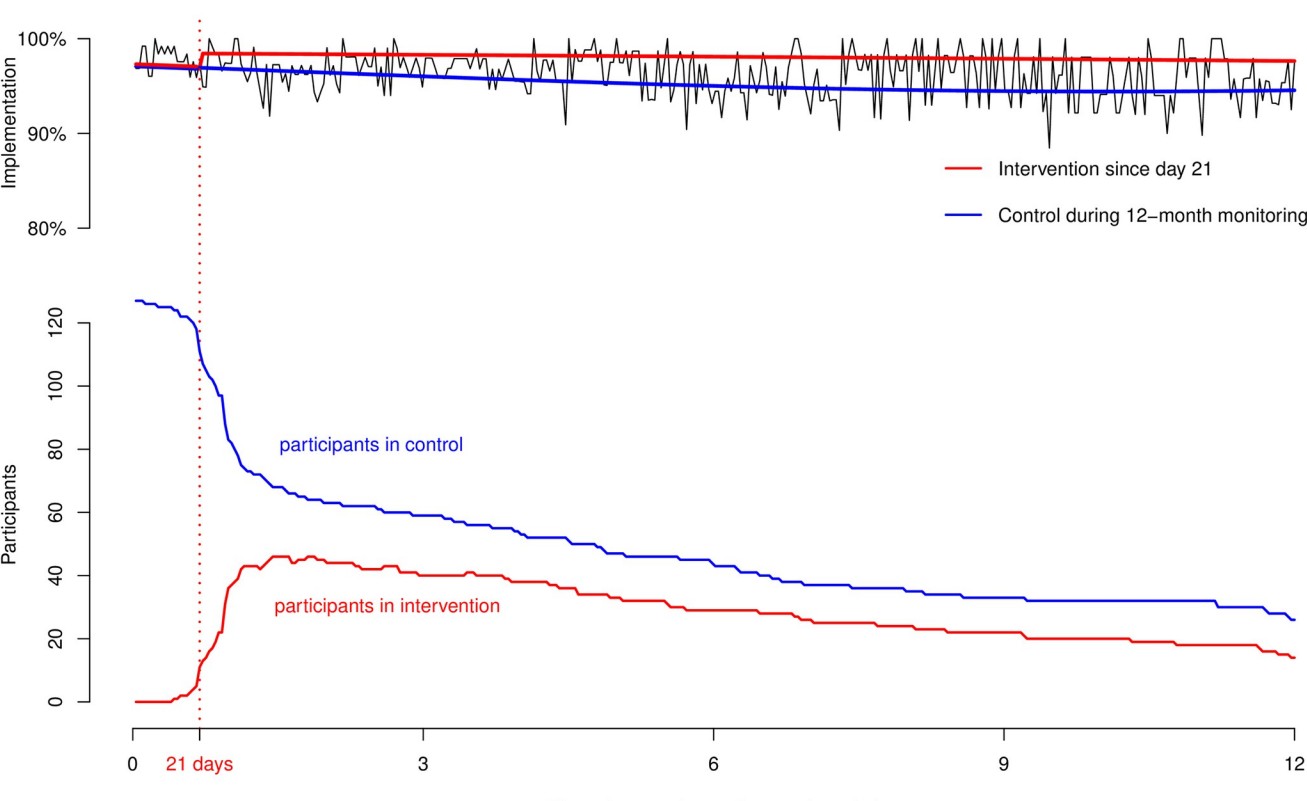

**Fig 2. GEE model showing PKI implementation during the 12-month monitoring period for a representative patient randomized in the intervention group since day 21 (red line) versus a representative control patient who stayed in the control group at randomization (blue line).** NB: the black lines on the background show empirical implementation in the total sample. The red and blue lines on the bottom of the figure represent the number of participants over time in the intervention and control group respectively.

at inclusion and 6 and 12 months post-inclusion and for patients who refused to participate, which indicated relatively high negative beliefs.

## IV. Discussion

### 1. Main results

The pharmacist-led IMAP improved statistically significantly patient implementation of PKIs during the 12-month monitoring period, whereas no impact was found on persistence or adherence or on BAM or QoL. Regarding medication implementation, the intervention benefited mostly men, patients younger than 60 years, patients prescribed PKIs longer than 60 days, patients without a diagnosis of metastasis or with a metastatic disease experience longer than 2 years, and patients who had never used any adherence tool in their therapeutic itinerary. As oncologists often prescribed transient interruptions in PKIs primarily to help patients recover from adverse drug events—according to evidence-based established guidelines and other administrative reasons—such adaptations needed to be considered in the analysis to avoid underestimating implementation and thus adherence.

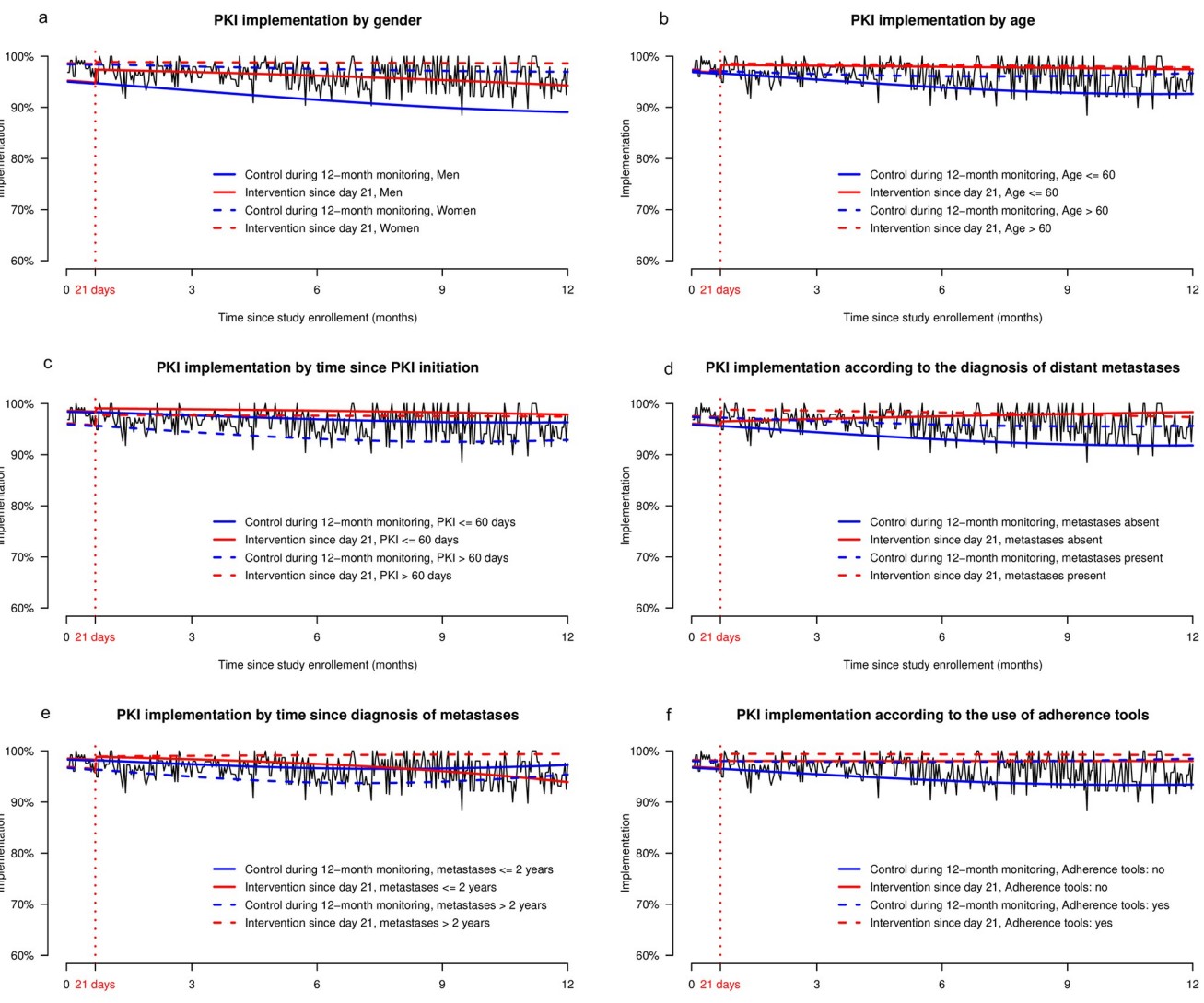

**Fig 3.** GEE models showing PKI implementation during the 12-month monitoring period for two representative patients randomized in the intervention group since day 21 (continued and dotted red lines) versus two representative control patients who stayed in the control group at randomization (continued and dotted blue lines); a: PKI implementation according to gender, b: according to patients' age, c: according to the time since PKI initiation, d: according to the diagnosis of distant metastases, e: according to the time since diagnosis of distant metastases, f: according to the use of adherence tools. NB: the black lines on the background show empirical implementation in the total sample.

## 2. Clinical outcomes, quality of life and beliefs about PKIs

While the IMAP increased treatment implementation by 3.1% points at 6 months in the intervention group compared to the control group, the impact of such an improvement on clinical outcomes (e.g., progression-free survival, tumor size) needs to be further investigated. In the OpTAT study, we were not able to perform such an analysis with a robust methodology because of numerous confounders (e.g., time since cancer diagnosis, time since PKI initiation, PKI prescription objective, PKI palliative lines, concomitant oncologic treatment). Future multicentric studies should consider monitoring medication adherence post-intervention along with progression-free survival and mortality in the cohort of included patients 1 to 5

**Table 2. Probabilities of PKI implementation in representative patients randomized in the intervention and control groups estimated by the GEE models.**

| At 6 months | Entire sample (%) | | | 95%CI of the difference (%) | |
|---|---|---|---|---|---|
| Global implementation to prescribed regimen | 96.27 | | | 94.46 | 97.5 |
| Global implementation according to usual recommended regimen by pharmaceutical industries | 88.53 | | | 84.19 | 91.8 |
| **Medication Implementation by Covariables** | | | | | |
| At 6 months | Intervention (%) | Control (%) | Difference (Δ%) | 95%CI of the difference (%) | |
| Randomization groups | 98.10 | 95.03 | 3.07 | 2.48 | 3.69 |
| In men | 96.24 | 91.52 | 4.71 | 3.38 | 6.00 |
| In women | 98.76 | 97.54 | 1.22 | 0.76 | 1.69 |
| Exclusion of female breast and gynecological cancers: In men | 96.12 | 91.97 | 4.15 | 2.79 | 5.48 |
| In women | 98.39 | 95.83 | 2.56 | 1.57 | 3.58 |
| In patients ≤ 60 years old | 97.94 | 93.94 | 4.00 | 3.17 | 4.85 |
| In patients > 60 years old | 98.28 | 96.12 | 2.17 | 1.50 | 2.84 |
| In patients with a time since diagnosis of metastases ≤ 2y | 97.48 | 96.63 | 0.85 | 0.06 | 1.64 |
| In patients with a time since diagnosis of metastases > 2y | 99.16 | 93.84 | 5.32 | 4.33 | 6.40 |
| In patients who had never used any adherence tool | 98.03 | 94.24 | 3.79 | 3.08 | 4.52 |
| In patients who had used adherence tool | 99.32 | 97.83 | 1.50 | 0.82 | 2.19 |
| In patients without a diagnosis of distant metastasis | 97.51 | 93.00 | 4.51 | 3.36 | 5.65 |
| In patients diagnosed with a diagnosis of distant metastasis | 98.27 | 95.91 | 2.36 | 1.77 | 2.97 |
| In patients whose time since PKI initiation was ≤ 60 days | 98.62 | 96.94 | 1.68 | 1.10 | 2.27 |
| > 60 days | 97.63 | 93.14 | 4.48 | 3.61 | 5.39 |
| **Medication Adherence** | | | | | |
| Adherence since randomization | 88.96 | 86.01 | 2.83 | -10.06 | 15.02 |
| **Medication Persistence** | | | | | |
| Persistence since randomisation | 91.52 | 89.26 | 2.26 | -9.98 | 14.50 |

years after the intervention. Indeed, such data are missing in the literature [8]. The impact of the intervention on patient-reported outcomes and experiences should also be further explored during and after the intervention. Our population reported low symptomatology and a high level of functioning in both groups after study inclusion, but the perceived global health status was approximately 6/10 (i.e., showing relatively low perceived QoL). Patients expressed an unfavorable balance between PKI harm versus necessity, which was not reversed during the intervention.

## 3. Determinants of implementation

In our study, implementation varied according to covariables. First, control patients without a diagnosis of metastasis who were prescribed a PKI as neo-adjuvant or adjuvant treatment implemented their PKI less; the perceived disease severity may have been lower among these patients than among patients with a diagnosis of metastasis. As previously reported in the literature and confirmed by our results, a longer treatment and metastatic disease experience may be a factor for suboptimal PKI implementation [16, 17]. We also showed that patients who had never used any tool to support medication adherence implemented their PKIs less in the control group. This suggests that past patient involvement in their care is a determinant of actual

**PKI implementation according to the consideration of alternate regimen versus usual recommended regimens from the pharmaceutical industry in the analysis**

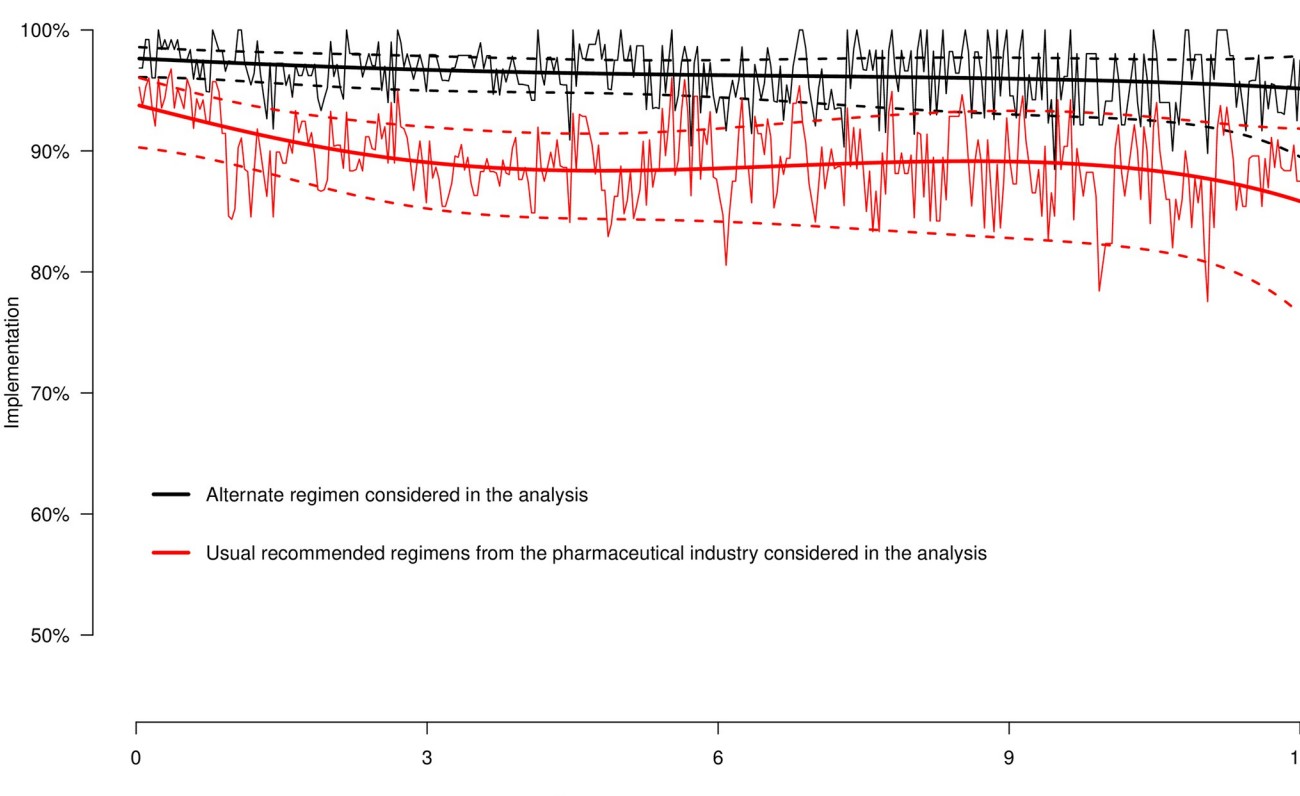

**Fig 4. GEE model showing PKI implementation during the 12-month monitoring period in the total sample, considering alternate regimens prescriptions (i.e., following oncologists' prescriptions, black lines) and on-label prescriptions (i.e., the usual recommended regimens from the pharmaceutical industry, red lines) and its 95%CI (dotted black and red lines).** NB: the black and red lines on the background show empirical implementation in the total sample considering alternate regimens (black lines) or on-label (red lines) prescriptions.

PKI implementation that is worth scrutinizing at the start of the IMAP intervention as an important lever for the intervention. Regarding the impact of gender on implementation, an observational study conducted at Lausanne University Hospital and the community pharmacy of Unisanté previously reported the same results, namely, that implementation of OATs decreased among men during a 12-month study [38]. Overall data on the impact of gender on medication adherence seem to be conflicting and vary according to age [41–43]. Gender is a potential confounder of the effects of an adherence intervention, which needs consideration in the design and analysis of future adherence studies [44].

## 4. Adverse events impact persistence in PKI use and interruptions in treatment

In our study, similarly in each group, 20% to 26% of patients discontinued their PKIs because of adverse events (i.e., 7/34 in the intervention and 6/23 in the control group). Oncologists prescribed repeated and transient PKI interruptions so that patients could recover from adverse events. If such interruptions were not prescribed, patients would certainly struggle to implement their PKIs optimally. Our results showed that the PKI dosing recommendation provided

by pharmaceutical industries often leads to clinically significant adverse events and toxicity in routine care. A systematic review also reported that alterations in PKI regimens such as alternate-day dosing, dose reductions or repeated dose interruptions are common with the prescription of OATs, and the impact on clinical outcomes should be further investigated [45], as well as on patient anxiety. In addition, to refine usual PKI dosage recommendations, rigor in monitoring adherence to OATs in oncology clinical trials with accurate measures should be reinforced (i.e., EM instead of pill count), as only 20% of trials reported adherence to OATs in their documentation for market authorization [46, 47]. In addition, to better adapt the PKI regimen to individual needs, a larger panel of dose strengths should be marketed along with systematic collection of patient-reported experience and outcomes [5] and routine pharmacokinetic/pharmacodynamic profiling [48]. As already noted in the literature [20], training community pharmacists on PKI objectives, lines of treatment, alternate regimens, adverse events and toxicity related to PKIs and drug–drug interactions are paramount to better inform patients and support them with their medication management.

## 5. Limitations and strengths of this study

The OpTAT study has strengths. First, the OpTAT study explored adherence to PKIs among patients with solid cancers, a vulnerable population of patients that is underinvestigated in the adherence literature. Our population included patients treated with a diversity of PKIs for common solid cancers. Second, the OpTAT study was implemented in a busy routine practice center as part of a semistructured intervention that has been implemented in routine care for other long-term diseases (e.g., HIV) for almost 30 years. Although the improvement in the intervention group versus the control group might be perceived as modest, it is important to note that medication implementation improved systematically on each single consecutive day over 12 months without exception. Third, we used the EM to objectively and longitudinally monitor two components of patient behavior toward medication management (i.e., implementation and persistence) over time. The rigorous cleaning and enrichment of the electronic adherence database allowed us to provide accurate measures of adherence in cancer care, where multiple treatment adjustments and transient interruptions are "the norm". We decreased the risk of misinterpreting and underestimating implementation and persistence in both groups. Fourth, statistical analysis performed on the EM adherence database was robust [22, 38], and the inclusion of covariables in the GEE models allowed us to detect relevant determinants explaining differential PKI implementation, hence deserving attention during a medication adherence intervention.

Some limitations are to be acknowledged. First, implementing an educational and behavioral intervention in a randomized controlled trial with the patient as the unit of randomization entails a risk of contamination of the control group, such that the control group will benefit indirectly from the intervention, resulting in an underestimation of the real impact of the intervention [49–51]. In the OpTAT study, the control group may have been polluted by the intervention group in several ways: i) control patients used the EM with the LCD screen, indicating the daily number of EM openings and time elapsed since the last opening, which helps prevent forgetfulness. We decided to do so to mimic some original PKI packaging, which provides tools such as daily or weekly blisters, in order not to deprive patients from existing marketed tools. ii) Pharmacists were specifically trained in oncology clinical pharmaceutical care, and the discussion at the counter with control patients may have been more tailored than in usual community pharmacies. iii) For ethical reasons, patients in the control group were called 72 hours after a missed appointment (versus immediately after the missed appointment among patients in the intervention group) to set another appointment for EM

refill, which does not happen in standard care because patient attendance at the pharmacy is not monitored. This process could have increased treatment persistence in control patients. iv) In the case of a cyclic regimen, the exact cycle dates were transmitted in writing to the patients in both groups and cross-checked with the patient and the oncologist in case of discrepancies. This professional attitude emerged from the pharmacists with the implementation of the OpTAT study, which might have increased adherence to PKIs with a cyclic regimen in both groups. v) Control patients were also asked to complete the questionnaires regarding their BAM and QoL at inclusion, 6 months and at study end, which could have led to specific thoughts indirectly impacting their adherence to PKIs. Second, the attrition rate was high in our study, which is aligned with the high attrition rates in oncology trials [52]. Indeed, numerous patients dropped out due to cancer progression, as expected and reported in the literature [53]. Other reasons for attrition were related to the burden of the study, which we aimed to alleviate. For example, to lower the patient burden in attending the intervention, pharmacy visits were scheduled after patients' medical appointments with the oncologist. Notably, some patients wanted to continue (intervention patients) or start (control patients) attending the IMAP after the end of this study, showing a real interest in long-term medication adherence support.

Third, patients underwent a baseline period before being randomized, but the first 45/130 (34.6%) patients were randomized at inclusion rather than after the baseline period (i.e., 23 patients in the intervention group and 22 in the control group), even if the intervention started after baseline for all patients. We cannot rule out that knowing they were included in the intervention group may have impacted medication adherence among these patients during the 3-week baseline period, although the intervention started only after baseline, based on the electronic adherence feedback. The visit at study inclusion was dedicated to explaining the study procedures and the use of the EM in a standardized way to each patient. Thus, for the 45 patients who were randomized at inclusion, the randomization date was considered the date of the first visit rather than the inclusion date. Finally, while our study showed no impact of the intervention on patients' BAM or QoL, a significant number of missing data (i.e., nonresponders to questionnaires) limited the interpretation of the results. However, patients were frequently asked to complete the questionnaires (i.e., the questionnaires were sent by mail, and reminders were provided by phone calls and during the pharmacy visits).

## V. Conclusions

This pharmacist-led interprofessional medication adherence program consistently improved patient implementation of PKIs over 12 months. The increase in patient implementation between groups might be seen as marginal, as implementation was high in both groups, probably in part due to the pollution of the control group by the intervention. However, the increase was consistent on every single consecutive day of the 12-month intervention. Men, patients younger than 60, those who had never used any adherence tool before inclusion, patients with no diagnosis of metastases or patients diagnosed with metastases more than 2 years before inclusion and patients who initiated PKIs more than 60 days before inclusion benefited more from the intervention. Oncologists frequently prescribed transient interruptions in PKIs to help patients cope with adverse drug events, showing that the usual dosing recommendation from pharmaceutical industries often needs to be adapted in routine clinical practice.

Further pragmatic trials should continue to rigorously evaluate interprofessional interventions to support adherence to OATs in routine care and should help to define how and why interventions improve medication adherence and when and with what intensity the interventions should be delivered in the patient care itinerary to maximize impact and equity [19]. As

it is estimated that 15 years is necessary to implement evidence-based practices in cancer control [54], future research should design interventions with hybrid designs using implementation sciences to better translate such interventions in routine clinical care and in the community [55, 56] to influence cancer care practices [57].

## Supporting information

**S1 Appendix. Design of the adherence part of the OpTAT study.**
(DOCX)

**S2 Appendix. Main reasons for non-participation reported by 103/111 patients who refused to participate.**
(DOCX)

**S3 Appendix. PKI persistence and adherence in both groups since randomization.**
(DOCX)

**S4 Appendix. Questionnaire scores in patients included in the intervention versus control groups at 6- and 12-month post-inclusion.**
(DOCX)

## Acknowledgments

We sincerely thank all patients who accepted to participate in the study, and those who refused and accepted to complete the questionnaires. We thank the Department of Oncology and the clinical team (oncologists, nurses, clinical pharmacists) for their contribution to patient recruitment and the community pharmacy of *Unisanté* for the implementation of the OpTAT study. We thank Dr Cyril Jaksic (Division of Clinical Epidemiology, Geneva University Hospital) for the analysis of the data of the BMQ and EORTC-QLQ-C30 questionnaires.

## Author Contributions

**Conceptualization:** Carole Bandiera, Evelina Cardoso, Isabella Locatelli, Dorothea Wagner, Chantal Csajka, Marie Paule Schneider.

**Data curation:** Carole Bandiera, Isabella Locatelli, Marie Paule Schneider.

**Formal analysis:** Carole Bandiera, Isabella Locatelli, Marie Paule Schneider.

**Funding acquisition:** Marie Paule Schneider.

**Investigation:** Carole Bandiera, Evelina Cardoso, Khalil Zaman, Antonella Diciolla, Antonia Digklia, Athina Stravodimou, Valérie Cristina, Veronica Aedo-Lopez, Ana Dolcan, Apostolos Sarivalasis, Hasna Bouchaab, Jennifer Dotta-Celio, Solange Peters, Dorothea Wagner, Chantal Csajka, Marie Paule Schneider.

**Methodology:** Carole Bandiera, Evelina Cardoso, Isabella Locatelli, Jérôme Pasquier, Chantal Csajka, Marie Paule Schneider.

**Project administration:** Carole Bandiera, Evelina Cardoso, Khalil Zaman, Dorothea Wagner, Chantal Csajka, Marie Paule Schneider.

**Resources:** Carole Bandiera, Evelina Cardoso, Khalil Zaman, Antonella Diciolla, Antonia Digklia, Athina Stravodimou, Valérie Cristina, Veronica Aedo-Lopez, Ana Dolcan, Apostolos Sarivalasis, Hasna Bouchaab, Jérôme Pasquier, Jennifer Dotta-Celio, Solange Peters, Dorothea Wagner, Chantal Csajka, Marie Paule Schneider.

**Software:** Isabella Locatelli, Jérôme Pasquier.

**Supervision:** Dorothea Wagner, Chantal Csajka, Marie Paule Schneider.

**Validation:** Carole Bandiera, Isabella Locatelli, Dorothea Wagner, Chantal Csajka, Marie Paule Schneider.

**Visualization:** Isabella Locatelli.

**Writing – original draft:** Carole Bandiera, Marie Paule Schneider.

**Writing – review & editing:** Carole Bandiera, Evelina Cardoso, Isabella Locatelli, Khalil Zaman, Antonella Diciolla, Antonia Digklia, Athina Stravodimou, Valérie Cristina, Veronica Aedo-Lopez, Ana Dolcan, Apostolos Sarivalasis, Hasna Bouchaab, Jérôme Pasquier, Jennifer Dotta-Celio, Solange Peters, Dorothea Wagner, Chantal Csajka, Marie Paule Schneider.

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
