## [Decision Letter · Decision Letter 0]

27 Nov 2023

PONE-D-23-24800A pharmacist-led interprofessional medication adherence program improved adherence to oral anticancer therapies: The OpTAT randomized controlled trialPLOS ONE

Dear Dr. Bandiera,

Thank you for submitting your manuscript to PLOS ONE. After careful consideration, we feel that it has merit but does not fully meet PLOS ONE’s publication criteria as it currently stands. Therefore, we invite you to submit a revised version of the manuscript that addresses the points raised during the review process.

 The paper was assessed by two peer-reviewers and one statistician who raised several important points to be addressed by the authors. Please submit a point-by-point response to all reviewers' comments.

We look forward to receiving your revised manuscript.

Kind regards,

Mabel Aoun, MD, MPH

Academic Editor

PLOS ONE

3. We notice that your supplementary figures and tables (Appendix 1-4) are included in the manuscript file. Please remove them and upload them with the file type 'Supporting Information'. Please ensure that each Supporting Information file has a legend listed in the manuscript after the references list.

4. We note that the original protocol that you have uploaded as a Supporting Information file contains an institutional logo. As this logo is likely copyrighted, we ask that you please remove it from this file and upload an updated version upon resubmission.

5. We note that the original protocol file you uploaded contains a confidentiality notice indicating that the protocol may not be shared publicly or be published. Please note, however, that the PLOS Editorial Policy requires that the original protocol be published alongside your manuscript in the event of acceptance. Please note that should your paper be accepted, all content including the protocol will be published under the Creative Commons Attribution (CC BY) 4.0 license, which means that it will be freely available online, and any third party is permitted to access, download, copy, distribute, and use these materials in any way, even commercially, with proper attribution.

Therefore, we ask that you please seek permission from the study sponsor or body imposing the restriction on sharing this document to publish this protocol under CC BY 4.0 if your work is accepted. We kindly ask that you upload a formal statement signed by an institutional representative clarifying whether you will be able to comply with this policy. Additionally, please upload a clean copy of the protocol with the confidentiality notice (and any copyrighted institutional logos or signatures) removed.

Reviewers' comments:

Reviewer's Responses to Questions

**Comments to the Author**

1. Is the manuscript technically sound, and do the data support the conclusions?

Reviewer #1: Yes

Reviewer #2: Yes

Reviewer #3: Partly

2. Has the statistical analysis been performed appropriately and rigorously? 

Reviewer #1: Yes

Reviewer #2: I Don't Know

Reviewer #3: Yes

3. Have the authors made all data underlying the findings in their manuscript fully available?

Reviewer #1: No

Reviewer #2: Yes

Reviewer #3: Yes

4. Is the manuscript presented in an intelligible fashion and written in standard English?

Reviewer #1: Yes

Reviewer #2: Yes

Reviewer #3: Yes

5. Review Comments to the Author

Reviewer #1: Abstract

1. The control arm received standard care plus EM without intervention. All PKIs were delivered in electronic monitors (EMs).

→ The control arm received standard care plus electronic monitor (EM) without intervention. All PKIs were delivered in EMs.

Introduction

1. Why is it necessary to investigate the adherence of protein kinase inhibitors (PKIs) in the context of solid tumors when the current body of literature has focused on their adherence in chronic myeloid leukemia (CML)? What justifies the need for a distinct examination of PKI adherence in solid tumors, and why can’t the findings from CML studies be directly extrapolated? Furthermore, what insights have previous studies provided regarding PKI adherence in CML? Incorporating these aspects into the introduction would help elucidate the significance of this study.

Methods

1. Could you explain your process for identifying and reaching out to potential pharmacists to obtain informed consent?

2. Which theoretical framework guided the study design and the selection of variables?

3. Could you please elaborate on the intervention process and the content that was implemented during the study?

4. What was the effect size and power of the study? How did you calculate and make sure the study sample size was large enough to have sufficient power for statistical analyses? Please provide more information to address these issues.

Results

1. Why did the two groups exhibit such a huge disparity in the median time spent on adherence after enrollment?

Discussion

1. Regarding medication implementation, the intervention benefited mostly men, patients younger than 60 years, patients prescribed PKIs longer than 60 days, patients without a diagnosis of metastasis or with a metastatic disease experience longer than 2 years, and patients who had never used any adherence tool in their therapeutic itinerary.

→ What could be the potential explanations for the factors identified in relation to medication implementation? Do the study findings align with the existing literature? You need to address these issues in the discussion section.

2. First, the OpTAT study explored adherence to PKIs among patients with advanced solid cancers,…

→ Was the cancer stage used as a criterion for participant recruitment? Without such criteria, making statements about advanced solid cancers may not be justified.

Reviewer #2: The study is rigorous, clear and accurate, I would however reformulate the hypothesis. The statements below are not clear and prone to confusion:

- We hypothesized that PKI implementation would be significantly lower if the alternate regimens (i.e., transient interruptions of PKI) prescribed by oncologists were not considered in the calculation

- Persistence would be improved

- These patients would perceive fewer prejudices and concerns about taking PKIs

Reviewer #3: This is a randomized clinical trial to evaluate the impact of a pharmacist-led interprofessional medication adherence program (IMAP) on patient implementation (dosing history), persistence (time until premature cessation of the treatment) and adherence to 27 PKIs prescribed for various solid cancers, as well as the impact on patients’ beliefs about medicines (BAM) and quality of life (QoL). The study included 118 patients who were randomized 1:1 into two arms. The study concluded that The IMAP, led by pharmacists in the context of an interprofessional collaborative practice, supported adherence, specifically implementation, to PKIs among patients with solid cancers. I have some concerns on the study design and statistical analysis.

1. The authors need to provide the sample size calculation and power analysis to justify the design of the study.

2. There are multiple testing that have been conducted in the study. However, there is no description on multiple comparison adjustment to avoid the inflation of false positive.

3. In Figure 5, there is no methodology description on how to generate the curves of the empirical implementation.

4. The major comparison is between the intervention and control groups, suggest removing the 9 patients of the not randomized from Table 1.

6. PLOS authors have the option to publish the peer review history of their article (what does this mean?). If published, this will include your full peer review and any attached files.

Reviewer #1: No

Reviewer #2: **Yes: **Myriam Noelle Watfa

Reviewer #3: No

---

## [Author Response · Author response to Decision Letter 0]

8 Mar 2024

Please see the rebuttal letter with the responses to the academic editor and the reviewers uploaded in the portal. Thank you.

---

## [Editor Report · Decision Letter 1]

15 May 2024

A pharmacist-led interprofessional medication adherence program improved adherence to oral anticancer therapies: The OpTAT randomized controlled trial

PONE-D-23-24800R1

Dear Dr. Bandiera,

We’re pleased to inform you that your manuscript has been judged scientifically suitable for publication and will be formally accepted for publication once it meets all outstanding technical requirements.

Kind regards,

Mabel Aoun, MD, MPH

Academic Editor

PLOS ONE
---

## [Editor Report · Acceptance letter]

29 May 2024

PONE-D-23-24800R1 

PLOS ONE

Dear Dr. Bandiera, 

I'm pleased to inform you that your manuscript has been deemed suitable for publication in PLOS ONE. Congratulations! Your manuscript is now being handed over to our production team.

Kind regards, 

on behalf of

Dr. Mabel Aoun 

Academic Editor

PLOS ONE